# Kalman Filtering Attention for User Behavior Modeling in CTR Prediction

**Hu Liu, Jing Lu, Xiwei Zhao, Sulong Xu, Hao Peng, Yutong Liu,**
**Zehua Zhang, Jian Li, Junsheng Jin, Yongjun Bao, Weipeng Yan**
Business Growth BU, JD.com
{liuhu1,lvjing12,zhaoxiwei,xusulong,penghao5,liuyutong,
zhangzehua,lijian21,jinjunsheng1,baoyongjun,paul.yan}@jd.com

## Abstract

Click-through rate (CTR) prediction is one of the fundamental tasks for e-commerce search engines. As search becomes more personalized, it is necessary to capture the user interest from rich behavior data. Existing user behavior modeling algorithms develop different attention mechanisms to emphasize query-relevant behaviors and suppress irrelevant ones. Despite being extensively studied, these attentions still suffer from two limitations. First, conventional attentions mostly limit the attention field only to a single user's behaviors, which is not suitable in e-commerce where users often hunt for new demands that are irrelevant to any historical behaviors. Second, these attentions are usually biased towards frequent behaviors, which is unreasonable since high frequency does not necessarily indicate great importance. To tackle the two limitations, we propose a novel attention mechanism, termed Kalman Filtering Attention (KFAtt), that considers the weighted pooling in attention as a maximum a posteriori (MAP) estimation. By incorporating a priori, KFAtt resorts to global statistics when few user behaviors are relevant. Moreover, a frequency capping mechanism is incorporated to correct the bias towards frequent behaviors. Offline experiments on both benchmark and a 10 billion scale real production dataset, together with an Online A/B test, show that KFAtt outperforms all compared state-of-the-arts. KFAtt has been deployed in the ranking system of JD.com, one of the largest B2C e-commerce websites in China, serving the main traffic of hundreds of millions of active users.

## 1 Introduction

CTR prediction is one of the fundamental tasks for e-commerce search engines. In contrast to early systems which only consider query keywords, modern search engines have become more personalized with the goal to "understand exactly what you mean and give you exactly what you want" [21]. Consequently, user behavior modeling, i.e. extracting users' hidden interest from historical behaviors, has been considered as one of the key components in CTR prediction for e-commerce search engines.

Nowadays, a popular user behavior modeling strategy is to estimate a user's hidden interest using the weighted pooling over one's historical behaviors. And these pooling weights are calculated by various *attention* mechanisms to emphasize query-relevant behaviors and suppress query-irrelevant ones. Despite being extensively studied, existing attention mechanisms for user behavior modeling still suffer from two limitations:

- Conventional attentions mostly assume that a user's interest under the current query keywords must be covered by one's historical behaviors. This assumption however, usually

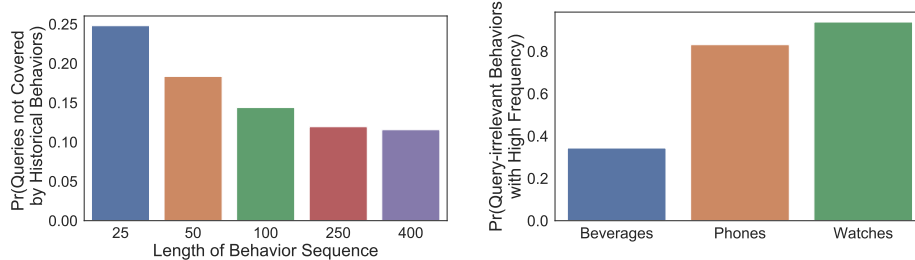

Figure 1: **Left**: Even when considering very long behavior sequences (length = 400), there is still a considerable fraction of queries irrelevant to any historical behaviors. **Right**: For a large fraction of queries, irrelevant behaviors are with high frequency and thus overwhelm the relevant behaviors. This fraction is extremely high in categories with low inherent frequency (93% for watches). Statistics are from 10 billion real search logs from JD.com.

      does not hold in the e-commerce scenarios, where users often hunt for *new demands* that are irrelevant to any history behaviors (Fig 1, left). In such case, attention only on historical behaviors, no matter how the pooling weights are allocated, mostly deviates from the real user interest and will thus mislead the CTR prediction system.

- Conventional attentions treat all historical behaviors independently, regardless of the hierarchical relationship between behaviors and their corresponding queries. More clearly, behaviors under the same query are highly homogeneous but make duplicated contribution in the weighted pooling. This certainly biases the attention weights towards frequent queries, which is unreasonable since high frequency does not necessarily indicate great importance. Given the huge variance in queries' *inherent frequency*, this bias becomes even more severe. An irrelevant but frequent query would easily overwhelm any closely related but infrequent one, and finally degrades the CTR prediction (Fig 1, right).

To address the first limitation, we propose Kalman Filtering Attention (KFAtt-base) that extends the attention field beyond the behaviors of one single user. Our algorithm is inspired by Kalman filtering [14] which has been wildly used in control theorem to estimate unobservable variables using a series of measurements. Specially, the historical behaviors could be modeled as measurements of the hidden user interest, each with different degrees of uncertainty. We formulate the estimation of the hidden user interest as MAP and provide a simple yet effective close-form solution. Compared to conventional attentions, this solution contains an additional global prior that enables unbiased hidden interest prediction even when few historical behaviors are relevant.

We further tackle the second limitation by proposing KFAtt-freq, an extension to KFAtt-base that captures the homogeneity of behaviors under the same query. In contrast to KFAtt-base that regards each behavior as an independent measurement, KFAtt-freq models each deduplicated query as a sensor and the behaviors under this query as repeated measurements from this sensor. We formulate this hidden interest estimation as MAP and derive the close-form solution. Compared to conventional attentions as well as KFAtt-base, this solution caps the total weights of behaviors under the same query and thus corrects the attention bias towards frequent queries.

Finally, for industrial scale online applications, we propose a KFAtt based behavior modeling module that incorporates many techniques to model behavior correlations and meet the online latency requirements. This module consists of two parts: A transformer based *encoder* for capturing correlations and dynamics of long behavior sequences and a KFAtt based *decoder* for extracting users' target-specific hidden interest.

The main contributions of this paper can be summarized as:

- To the best of our knowledge, we are the first to high-light the two limitations in conventional attentions for user behavior modeling, namely, the limited attention field on a single user's behavior and the attention bias towards frequent behaviors.
- We propose a novel attention mechanism KFAtt that successfully addresses the two limitations and validate it through concrete theoretical analysis and extensive experiments, both offline & Online A/B test, demonstrating our validity, effectiveness and adaptability.

- Based on KFAtt, we propose an efficient behavior modeling module that meets the strict online latency requirements, and deploy it in the search engine of JD.com, one of the largest B2C e-commerce websites in China. KFAtt is now serving the main traffic of hundreds of millions of active users.

## 2 Related Work

CTR prediction, which aims to predict the probability that a user clicks an ad, is one of the fundamental tasks in the online advertising industry. Pioneer CTR models are usually linear [22], collaborative filtering-based [23] or tree-based [12]. With the rise of deep learning, most recent CTR models share an Embedding and Multi-layer Perceptron (MLP) paradigm [4, 3, 17]. Based on this paradigm, polynomial networks are introduced for feature interactions [11, 27].

User behaviors modeling is a crucial component in CTR prediction, which usually extracts vast and insightful information about user interest [15, 16, 2, 7]. Limited by computational resources, early works are mostly in target-independent manners [6, 24, 26] which are super efficient or even could be calculated offline. Since they only extract users' general interest, not interest in specific queries or items, their contribution to the CTR prediction is mostly limited. Recently, various attention mechanisms are adopted in user behavior modeling to extract target-dependent interest [31, 9, 29]. By focusing on target-relevant behaviors, these algorithms achieve state-of-the-art performance in CTR prediction. Despite their great success, most recent behavior modeling algorithms focus on applying attention to different network structures, i.e. RNN [30], Memory Network [20] and Transformer [29]. While to the best of our knowledge, few are dedicated to addressing the limitations of attention mechanisms themselves.

For attention mechanisms out of CTR / behavior modeling, the idea of not assuming target in the input sequences appears in [13]. While they only depicts the uncertainty, we make an unbiased estimation by incorporating a priori. The bias towards frequent behaviors is addressed in [28] by incorporating global inverse word frequency. We instead address the frequency variance within a user's behaviors, and prevent the attention output from being overwhelmed by irrelevant but frequent behaviors.

## 3 Method

We first review the background of user behavior modeling with the contexts of CTR prediction. Then we introduce **KFAtt**, our attention mechanism specially designed for behavior modeling. Finally, we adopt KFAtt to the behavior modeling module in the real online CTR prediction system.

### 3.1 Preliminaries

CTR prediction, to predict the probability that a *user* clicks an *item*, is one of the fundamental tasks in search engines in e-commerce industry. A CTR prediction model mostly takes five fields of features:

$$\text{CTR} = f(\text{query}, \text{user behaviors}, \text{user profile}, \text{item profile}, \text{contexts}). \qquad (1)$$

Among them, user behaviors faithfully reflect users' immediate and evolving interest and sometimes even disclose one's future clicks. Consequently, user behavior modeling has been considered as a key component in the CTR prediction task [19]. A behavior modeling module is usually formulated as:

$$\hat{\mathbf{v}}_q = \text{User-Behavior}(\mathbf{q}, \mathbf{k}_{1:T}, \mathbf{v}_{1:T}) \qquad (2)$$

Namely, the aim is to predict the user's hidden interest $\hat{\mathbf{v}}_q$ under the current query $\mathbf{q}$, given $T$ historical clicked items $\mathbf{v}_{1:T}$, together with their corresponding query words $\mathbf{k}_{1:T}$. [1]

In literature, a commonly used behavior modeling strategy is to adopt an attention mechanism over the user's historically clicked items, i.e., $\hat{\mathbf{v}}_q = \sum_{t=1}^{T} \alpha_t \mathbf{v}_t$, where, $\alpha_t > 0$ is the combination weight learnt from attention. An intuitive idea is to focus on the clicks under similar queries,

$$\alpha_t = \frac{\exp(\mathbf{q}^\top \mathbf{k}_t)}{\sum_{\tau=1}^{T} \exp(\mathbf{q}^\top \mathbf{k}_\tau)} \qquad (3)$$

Advanced attention mechanisms to learn $\alpha$ include DIN [31], DSIN [30] among others.

Despite being extensively studied, most of the existing attention mechanisms adopted in user behavior modeling still suffer from two limitations: 1). the limited attention field only on a single user's historical behaviors that often cannot cover the current interest, 2). and the bias in attention weights towards frequent behaviors. As a result, the predicted hidden interest $\hat{\mathbf{v}}_q$ usually deviates from the real user interest and finally degrades the CTR prediction system.

## 3.2 Kalman Filtering Attention for User Behavior Modeling

We address the first limitation by proposing a novel attention mechanism, Kalman Filtering Attention (KFAtt), that extends the attention field beyond the historical behaviors of one single user.

Our algorithm is inspired by the Kalman filtering [14], which has been wildly used in control theorem to estimate unobservable variables using a series of sensors. We now reformulate user behavior modeling in the problem setting of Kalman filtering.

The aim is to estimate an unobservable variable $\mathbf{v}_q$, i.e. the user's interest given the current query $\mathbf{q}$. We assume that $\mathbf{v}_q$ follows a Gaussian distribution, $\mathbf{v}_q \sim \mathcal{N}(\boldsymbol{\mu}_q, \sigma_q^2 I)$. This randomness characterizes the divergent interests of a great many users under the same query. Specifically, $\boldsymbol{\mu}_q$ represents the mean interest under the same $\mathbf{q}$ across all users. And $\sigma_q$ represents the interest diversity across all users, which is an inherent attribute of queries. For example, $\sigma_q$ for query "new year gift" is large, while $\sigma_q$ for "Nike running shoes" is small. In practice, both $\boldsymbol{\mu}_q$ and $\sigma_q$ can be calculated from $\mathbf{q}$ using 2-layer MLP's trained jointly with the CTR model.

We model each click as a measurement of the hidden interest, and the corresponding query as the sensor for this measurement. Namely, Kalman filtering measures $\mathbf{v}_q$ by $T$ *unbiased* sensors $\mathbf{k}_{1:T}$ and gets a series of measurements, $\mathbf{v}_{1:T}$, each with different degree of uncertainty. These measurements are assumed to follow Gaussian distributions conditioning on the measured variable,

$$\mathbf{v}_t | \mathbf{v}_q \sim \mathcal{N}(\mathbf{v}_q, \sigma_t^2 I), t \in \{1, ..., T\} \tag{4}$$

The uncertainty $\sigma_t$ depends on the distance between the sensor $\mathbf{k}_t$ and the measured variable $\mathbf{q}$, or namely, the distance between the current query and the historical query.

We now estimate the hidden variable $\mathbf{v}_q$ using the maximum a posteriori (MAP) criterion:

$$\hat{\mathbf{v}}_q = \underset{\mathbf{v}_q}{\operatorname{argmax}}\, p(\mathbf{v}_q) \prod_{t=1}^{T} p(\mathbf{v}_t | \mathbf{v}_q) = \underset{\mathbf{v}_q}{\operatorname{argmax}}\, \varphi(\mathbf{v}_q | \boldsymbol{\mu}_q, \sigma_q^2 I) \prod_{t=1}^{T} \varphi(\mathbf{v}_t | \mathbf{v}_q, \sigma_t^2 I) \tag{5}$$

where $\varphi$ represents the Gaussian PDF. This optimization enjoys an easy closed form solution,[2]

$$\hat{\mathbf{v}}_q(\mathbf{q}, \mathbf{k}_{1:T}, \mathbf{v}_{1:T}) = \frac{\frac{1}{\sigma_q^2} \boldsymbol{\mu}_q + \sum_{t=1}^{T} \frac{1}{\sigma_t^2} \mathbf{v}_t}{\frac{1}{\sigma_q^2} + \sum_{t=1}^{T} \frac{1}{\sigma_t^2}} \tag{6}$$

**Remarks**: Apart from the historical clicks $\mathbf{v}_{1:T}$ used in conventional attentions, our solution also incorporates the global prior $\boldsymbol{\mu}_q$. For a new query with few close-related historical clicks, all $\sigma_t$'s are large. Our solution automatically resorts to the global mean $\boldsymbol{\mu}_q$, i.e. what most other users click under this query. We now highlight our first advantage over conventional attentions: by incorporating the global prior, we never restrict the attention field to behaviors of a single user, but are able to make unbiased hidden interest prediction even when few relevant behaviors are available.

In addition, when setting $\sigma_q = \infty$ and $1/\sigma_t^2 = \exp(\mathbf{q}^\top \mathbf{k}_t)$, i.e. neglecting the global prior, Eq (6) degenerates to the traditional attention in Eq (3). This observation supports not only the validity of KFAtt, but also the flexibility. In general, we can easily adopt KFAtt to improve any existing attention mechanisms, by just assigning $1/\sigma_t^2$ to their own attention weights. We highlight this as our second advantage and will discuss more in Section 3.4 and experiments.

## 3.3 Kalman Filtering Attention with Frequency Capping

We extend KFAtt to address the 2nd limitation of conventional attentions, i.e. the bias towards frequent behaviors. The main idea is to correct the bias through capturing the homogeneity of behaviors under the same query. We term this extension as **KFAtt-freq** and the basic one as **KFAtt-base**.

While KFAtt-base models all the historical queries as independent sensors, $\mathbf{k}_{1:T}$ actually contain many duplications of frequent queries. So in KFAtt-freq, only deduplicated queries are modeled as sensors. Formally, KFatt-freq measures $\mathbf{v}_q$ using a series of independent sensors $\mathbf{k}_{1:M}$, where $M \leq T$ is the number of deduplicated queries. On sensor $\mathbf{k}_m$, we get $n_m$ observations $[\mathbf{v}_{m,1}, ..., \mathbf{v}_{m,n_m}]$, each corresponding to a click under query $\mathbf{k}_m$. Obviously, $\sum_{m=1}^{M} n_m = T$.

The observational error in $\mathbf{v}_{m,t}$ can be decomposed into two independent parts: 1). the *system error* $\sigma_m$ that results from the distance between the measured variable $\mathbf{v}_q$ and the sensor $\mathbf{k}_m$, 2). and the *random error* $\sigma'_m$ that naturally lies in the multiple observations on sensor $\mathbf{k}_m$. In practice, $\sigma'_m$ can be calculated from $\mathbf{k}_m$ using a 2-layer MLP trained jointly with the CTR model.

To model the system error, we introduce the first Gaussian distribution,

$$\mathbf{v}_m | \mathbf{v}_q \sim \mathcal{N}(\mathbf{v}_q, \sigma_m^2 I), m \in \{1, ..., M\} \tag{7}$$

where $\mathbf{v}_m$ denotes the value on sensor $\mathbf{k}_m$ excluding the random error. And to model the random error, we introduce the second Gaussian distribution,

$$\mathbf{v}_{m,t} | \mathbf{v}_m \sim \mathcal{N}(\mathbf{v}_m, \sigma_m'^2 I), t \in \{1, ..., n_m\} \tag{8}$$

So far $\mathbf{v}_q$ can be estimated by the maximum a posteriori criterion:

$$
\begin{aligned}
\hat{\mathbf{v}}_q &= \operatorname*{argmax}_{\mathbf{v}_q} p(\mathbf{v}_q) \prod_{m=1}^{M} \left[ p(\mathbf{v}_m | \mathbf{v}_q) \prod_{t=1}^{n_m} p(\mathbf{v}_{m,t} | \mathbf{v}_m) \right] \\
&= \operatorname*{argmax}_{\mathbf{v}_q} \varphi(\mathbf{v}_q | \boldsymbol{\mu}_q, \sigma_q^2 I) \prod_{m=1}^{M} \left[ \varphi(\mathbf{v}_m | \mathbf{v}_q, \sigma_m^2 I) \prod_{t=1}^{n_m} \varphi(\mathbf{v}_{m,t} | \mathbf{v}_m, \sigma_m'^2 I) \right]
\end{aligned}
\tag{9}
$$

This optimization enjoys a closed form solution,

$$\hat{\mathbf{v}}_q(\mathbf{q}, (\mathbf{k}_m, \mathbf{v}_{m,1:n_m})_{m=1:M}) = \frac{\frac{1}{\sigma_q^2}\boldsymbol{\mu}_q + \sum_{m=1}^{M} \frac{1}{\sigma_m^2 + \sigma_m'^2/n_m} \overline{\mathbf{v}}_m}{\frac{1}{\sigma_q^2} + \sum_{m=1}^{M} \frac{1}{\sigma_m^2 + \sigma_m'^2/n_m}} \tag{10}$$

where $\overline{\mathbf{v}}_m = \frac{1}{n_m} \sum_{t=1}^{n_m} \mathbf{v}_{m,t}$ is the mean over all observations on sensor $\mathbf{k}_m$.

**Remarks**: As indicated by $\sigma_m$, the weight of a behavior is still related to its distance to the current query. While different from KFAtt-base, this weight does not increase linearly with the behavior frequency. For an irrelevant behavior (large $\sigma_m$), even assuming frequency $n_m \to \infty$, its weight $\frac{1}{\sigma_m^2}$ is still neglectable. We now highlight our advantage over conventional attentions as well as KFAtt-base: we cap the total weight of behaviors under the same query and thus correct the bias towards frequent queries in user interest prediction, which will further contribute to the CTR prediction task.

In addition, when $n_m = 1$ and $\sigma'_m = 0$, i.e. assuming each query is associated with only one click, Eq (10) degenerates into KFAtt-base, which supports the validity of KFatt-freq.

Finally, KFAtt-freq also enjoys similar flexibility to KFAtt-base. We can easily adopt KFatt-freq to improve any existing attentions, by adjusting $\sigma_m$ according to their own attention weights.

## 3.4 Kalman Filtering Attention in Real Online System

Previously, we focused purely on the attention mechanism. While industrial scale behavior modeling actually includes many techniques to precisely extract user interest and meet the low latency requirements of online systems. We now introduce the whole behavior modeling module deployed in our online CTR prediction system, which consists of two parts: a transformer based [25] encoder that captures the correlations and dynamics of behaviors, and a KFAtt based decoder to predict query-specific user interest. We term the whole module **KFAtt-trans**.

### 3.4.1 Encoder: Within Session Interest Extractor

To model the sequential order of behaviors, we inject a position encoding [10] to $\mathbf{k}_{1:T}, \mathbf{v}_{1:T}$. And to capture the correlation between behaviors, we adopt the multi-head self-attention used in Transformer [25]. While differently, considering the very long behavior sequences in industry, this self-attention is only conducted locally for efficiency.

Specially, we divide the behavior sequence into *Sessions* according to their occurring time. Since the inter-session correlation is usually small [8], the self-attention is only conducted within sessions. We denote the behaviors in session $s$ as $K_s, V_s \in \mathbb{R}^{T_s \times d_{\text{model}}}$, where $T_s$ is the number of behaviors in session $s$, and each row in matrix $K_s$ / $V_s$ is a historical query / click. The self-attention is:

$$\text{MultiHead}(K_s, K_s, V_s) = \text{Concat}(\text{head}_1, \dots, \text{head}_h)W^O$$

$$\text{head}_i = \text{Attention}(K_s W_i^Q, K_s W_i^K, V_s W_i^V) = \text{softmax}(K_s W_i^Q W_i^{K^\top} K_s^\top / \sqrt{d_k})V_s W_i^V \quad (11)$$

where $W_i^Q, W_i^K \in \mathbb{R}^{d_{\text{model}} \times d_k}$, $W_i^V \in \mathbb{R}^{d_{\text{model}} \times d_v}$, and $W^O \in \mathbb{R}^{h d_v \times d_{\text{model}}}$ are projection matrices. The output of the self-attention is then processed by a FC layer to generate the session interest $H_s \in \mathbb{R}^{T_s \times d_{model}}$, where each row corresponds to one behavior refined by the local correlation.

### 3.4.2 Decoder: Query-specific Interest Aggregator

As discussed previously, KFAtt enjoys the flexibility to be adopted to any attentive model. The only adaption needed is to adjust the system error $1/\sigma^2$ according to the distance metric. Now KFAtt acts as the decoder to aggregate interest from all sessions for query-specific interest prediction,

$$\hat{\mathbf{v}}_q = \text{Concat}(\text{head}_1, \dots, \text{head}_h)W^O, \quad \text{head}_i = \text{KFAtt}(\mathbf{q}^\top W_i^Q, K W_i^K, H W_i^V) \quad (12)$$

where $K, H \in \mathbb{R}^{T \times d_{model}}$ are gathered from $K_s, H_s$ across all sessions and KFAtt stands for the solution in Eq (6) or (10) with the system error $1/\sigma_t^2$ or $1/\sigma_m^2$ set to $\exp(\mathbf{q}^\top W_i^Q W_i^{K^\top}\mathbf{k})$.

### 3.5 Why Kalman Filtering?

KF is essentially a sensor-fusion method. The fusion is estimated by MAP, whose solution is a weighted-sum of *prior* and sensor measurements. Similarly in behavior modeling, each historical behavior can be considered as a measurement of the current interest. So the estimated current interest is also a fusion, which is naturally under MAP framework and thus fits KF. While conventional attentions neglect query priors and thus suffer from cold start.

## 4 Experiments

Our experiments are organized into two groups:

(i) To exam the effectiveness of our behavior modeling module, we compare KFAtt-trans to many state-of-the-arts on a wildly used benchmark dataset. And to validate the adaptability of the proposed attention mechanism, we plugin KFAtt to various attentions and show the consistent improvement.

(ii) We further exam the contribution of KFAtt to the whole CTR prediction system. Experiments include A). offline evaluations on JD's ten-billion scale real production dataset, and B). an online A/B test on the real traffic of hundreds of millions of active users on JD.com.

### 4.1 Dataset and Evaluation Metrics

**Amazon Dataset** [18] is a commonly used benchmark in user behavior modeling [31, 30]. We use the 5-core Electronics subset, including 1,689,188 instances with 192,403 users and 63,001 goods from 801 categories. The task is to predict whether a user will write a review for a target item given historical reviews. Here, the reviewed item is regarded as behavior $\mathbf{v}$, the category of target item as $\mathbf{q}$ and the category of reviewed item as $\mathbf{k}$. Following [31], the last review of each user is used for testing and the others for training. Negative instances are randomly sampled from not reviewed items of this user. To focus on behavior modeling itself and eliminate interference from other fields, all compared algorithms discard other features except reviews.

**Real Production Dataset** is the traffic logs from the search advertising system of JD.com. 10 billion click-through logs in the first 32 days are used for training, and 0.5 million from the following day for testing. User clicks / queries in previous 70 days are used as behaviors $\mathbf{v}$ / $\mathbf{k}$, along with abundant multi-modal features including the query, user profile, ad profile, ad image and context.

**Evaluation Metric.** AUC is almost the default offline evaluation metric in the advertising industry since offline AUC directly reflects the online performance. We use AUC for all offline evaluation, both

Table 1: Comparison with state-of-the-arts (AUC). Mean over 5 runs with random initialization and instance permutations. Std ≈0.1%, extremely statistically *significant* under unpaired t-test.

| Amazon | Pooling | Vanilla | DIN | DIEN | Transformer | KFAtt-trans-b | KFAtt-trans-f |
|--------|---------|---------|--------|--------|-------------|---------------|---------------|
| All    | 0.7727  | 0.8034  | 0.8317 | 0.8684 | 0.8720      | 0.8766        | **0.8789**    |
| New    | 0.7555  | 0.7677  | 0.8038 | 0.8465 | 0.8488      | 0.8552        | **0.8578**    |
| Infreq | 0.7397  | 0.7596  | 0.7975 | 0.8381 | 0.8414      | 0.8465        | **0.8496**    |

Table 2: Adaptation to various attentions mechanisms (AUC).

| Data | Vanilla Att | | | DIN | | | Transformer | | |
|------|--------|---------|---------|--------|---------|---------|--------|---------|---------|
|      | Origin | KFAtt-b | KFAtt-f | Origin | KFAtt-b | KFAtt-f | Origin | KFAtt-b | KFAtt-f |
| All    | 0.8034 | 0.8457 | **0.8481** | 0.8317 | 0.8479 | **0.8524** | 0.8720 | 0.8766 | **0.8789** |
| New    | 0.7677 | 0.8174 | **0.8231** | 0.8038 | **0.8218** | 0.8214 | 0.8488 | 0.8552 | **0.8578** |
| Infreq | 0.7596 | 0.8067 | **0.8085** | 0.7975 | 0.8148 | **0.8159** | 0.8414 | 0.8465 | **0.8496** |

on benchmark and real production dataset. Specially, $\text{AUC} = \frac{1}{|\mathcal{D}^-||\mathcal{D}^+|} \sum_{i \in \mathcal{D}^-} \sum_{j \in \mathcal{D}^+} \mathbb{I}(\hat{y}_i < \hat{y}_j)$, where $\mathbb{I}$ is the indicator and $\mathcal{D}^-$ and $\mathcal{D}^+$ are the sets of negative and positive examples.

## 4.2 Compared Algorithms

We compare to state-of-the-art user behavior modeling algorithms including: **Pooling**: All user behaviors are treated equally with the sum pooling operation. **Vanilla Attention**: Attentive aggregate user behaviors, with $\alpha$ defined in Eq (3). **DIN** [31]: Attentive aggregate user behaviors with dedicated designed local activation unit. **DIEN** [30]: A GRU [5] encoder to capture the dynamics, followed by another GRU with attentional update gate to depict interest evolution.

## 4.3 Implementation Details

For ablation studies on Amazon, all algorithms are implemented in Tensorflow [1], based on the code of DIEN [3], following their parameter settings (learning rate, batch size, etc). Since the behavior sequence on Amazon is short, we regard the whole sequence as one session. For experiments on the real production dataset, all 96 multi-modal features are first embedded to 16-dimensional vectors and then processed by a 4-layer MLP with dimension 1024, 512, 256, 1. When there is a 30 minutes' time interval between adjacent behaviors, we conduct a session segmentation. For each instance, we use at most 10 sessions and 25 behaviors per session. The learnt hidden user interest, $\hat{\mathbf{v}}_q \in \mathbb{R}^{64}$ is concatenated to the output of $1_{st}$ FC layer together with a 150-dimensional visual feature vector.

## 4.4 Comparison with State-of-the-arts

We aim to show the performance gain from both the global prior and frequency capping. To highlight the two advantages, besides testing on "*All*" test instances, we also report the performance on 2 more challenging subsets: "*New*" where the current query **q** is irrelevant to any historical queries $\mathbf{k}_{1:T}$, and "*Infreq*" where **q** is from an infrequent category [4] and thus relevant behaviors would easily be overwhelmed by irrelevant but frequent ones. Performance is shown in Table 1.

Both KFAtt-trans-base and KFAtt-trans-freq outperform all state-of-the-arts, including Transformer whose only difference to the proposed algorithms lies in the attention mechanism. And on the two difficult subsets, KFAtt even achieves larger performance gain. By incorporating global prior and frequency capping, KFAtt successfully addresses the challenges of new and infrequent queries and thus is more suitable for behavior modeling than existing attentions.

Table 3: Experiments on production dataset.

| | | |
|---|---|---|
| Offline | Base (Sum pooling) | 0.749 |
| AUC | + DIN | 0.755 (**+0.006**) |
| (+ | + Transformer | 0.760 (**+0.011**) |
| AUC | + KFAtt-trans-b | 0.764 (**+0.015**) |
| Gain) | + KFAtt-trans-f | 0.766 (**+0.017**) |

| Online | CTRgain | CPCgain | eCPMgain |
|---|---|---|---|
| KFAtt-trans-f | +4.40% | -0.33% | +4.06% |

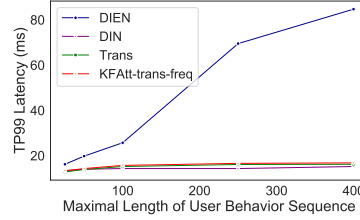

Figure 2: TP99 latency in real online CTR system w.r.t length of behavior sequences.

Moreover, we analyze AUC of compared algorithms. The gain from DIN to DIEN validates the significance of modeling the sequential pattern. And the gain from DIEN to Transformer supports the importance of self-attention in capturing behavior correlations. This further validates the design of our whole behavior modeling module, namely how to properly adopt KFAtt to real online system.

### 4.5 Adaptation to Various Attentions Mechanisms

Theoretically, KFAtt could be used to improve any attention mechanism by simply adjusting the system error. To validate the adaptability of KFAtt, we assign attention weights calculated by Vanilla, DIN and Transformer to $1/\sigma_t^2$ and $1/\sigma_m^2$ in Eq (6) and (10) and compare them with their original counterparts. Results are shown in Table 2. We observe that when plugging KFAtt, all the three attention mechanisms acquire consistent improvement, validating our strong adaptability.

### 4.6 Experiments on Real Production Dataset & Online A/B Testing

We exam the contribution of KFAtt-trans to the whole CTR prediction system in Table 3[5].

In offline experiments on the real production dataset, we observe significant improvement from advanced behavior modeling modules, though the base model in our ad system has already been highly optimized on 10 billion scale multi-modal data with hundreds of millions of vocabularies. This again supports the significance of behavior modeling. Empirically, new queries and frequency variance are more common in real e-commerce systems than academic benchmarks. This also contributes to the advantage of KFAtt over Transformer and other baselines.

In the online A/B test, KFAtt-freq contributes to 4.4% CTR gain compared to the base model (sum pooling). This is nontrivial given that all other components of JD's base model have already been highly optimized and this leads to an additional $0.1Billion/year ad income. To exam our efficiency, we plot the online latency of KFAtt in comparison to state-of-the-arts in Fig 2. The light-weighted module KFAtt enjoys the similar high efficiency as DIN and Transformer and outperforms DIEN.

## 5 Conclusions

We proposed a novel attention mechanism, termed Kalman Filtering Attention, which considered the weighted pooling in attention as maximum a posteriori estimation. KFAtt addresses the common limitations of existing attention mechanisms, namely, the limited attention field within a single user's behaviors and the bias towards frequent behaviors, which contributes to significant performance gain in the following CTR prediction tasks. Together with a highly efficient behavior modeling module, KFAtt has been deployed in the search engine of JD.com, serving the main traffic of hundreds of millions of active users everyday.

We believe that KFAtt is a widely applicable method that is not restricted to search scenarios. For example, in recommendation, query $\mathbf{q}$, $\mathbf{k}$ could be the item category that the user browses. We are excited about the future application of KFAtt to more attention mechanisms (e.g. self-attention, co-attention), and to more types of data (e.g. sequence, image and graph). Another interesting future direction is to change the point estimation of KFAtt to interval estimation (the confidence of attention output), which may help to depict the prediction reliability.

## Broader Impact

Ad-tech and e-commerce practitioners are the clearest immediate beneficiaries. Might have applications in other areas that use behaviors for personalized services as well. Also notice that the prior in KFAtt might not be useful for neural machine translation and question answering, since the target are always covered in the input sequence.

## Funding Disclosure

JD.com grants the production dataset, the online system and all computing resources for this work.

## Footnotes

[1]Note that our term *query* actually indicates a general setting, not limited to the key words in search scenario. For example, in recommendation, query could be the product category that the user is browsing. Or simpler, $\mathbf{k} = \mathbf{v}$ both represent the clicked item and $\mathbf{q}$ is the target item, which was used in DIN [31].

[2]Proofs of Eq (6) and (10) are in the supplementary materials.

[3] https://github.com/mouna99/dien/tree/1f314d16aa1700ee02777e6163fb8ca94e3d2810.

[4] Category frequency less than 2000 in the training set.

[5]We exclude DIEN here due to its high latency in our CPU-based online system (Fig 2).

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
