[Supplementary Material]

# Kalman Filtering Attention for User Behavior Modeling in CTR prediction

**Hu Liu, Jing Lu, Xiwei Zhao, Sulong Xu, Hao Peng, Yutong Liu,**
**Zehua Zhang, Jian Li, Junsheng Jin, Yongjun Bao, Weipeng Yan**
Business Growth BU, JD.com
{liuhu1,lvjing12,zhaoxiwei,xusulong,penghao5,liuyutong,
zhangzehua,lijian21,jinjunsheng1,baoyongjun,paul.yan}@jd.com

## 1 Notations

Table 1: Important Notations Used in Methods

| | | | |
|---|---|---|---|
| $\mathbf{q}$ | current query | $\hat{\mathbf{v}}_q$ | predicted interest under query $\mathbf{q}$ |
| $T$ | # historical behaviors | $\mathbf{k}$ | historical query / sensor |
| $\alpha$ | attention weight | $\mathbf{v}$ | historical click / measured value |
| $\boldsymbol{\mu}_q, \sigma_q$ | mean & std for query $\mathbf{q}$ | $\varphi$ | Gaussian probability density |
| $\sigma_t$ | std for query/sensor | $m, M$ | index of & # deduplicated queries |
| $t$ | index for action | $n_m$ | # clicks associated to query $\mathbf{k}_m$ |
| $\sigma_m$ | system error of sensor $\mathbf{k}_m$ | $\sigma'_m$ | random error of sensor $\mathbf{k}_m$ |

## 2 Proofs of KFAtt Solutions

### 2.1 KFAtt-base

To estimate the hidden variable $\mathbf{v}_q$ using the MAP criterion, the function to be maximized in KFAtt-base is given by:

$$
\begin{aligned}
F_{base}(\mathbf{v}_q) &= \varphi(\mathbf{v}_q|\boldsymbol{\mu}_q, \sigma_q^2 I)\prod_{t=1}^{T}\varphi(\mathbf{v}_t|\mathbf{v}_q, \sigma_t^2 I) \\
&= \frac{1}{\Sigma}\exp\left(-\frac{1}{2\sigma_q^2}(\mathbf{v}_q - \boldsymbol{\mu}_q)^\top(\mathbf{v}_q - \boldsymbol{\mu}_q) + \sum_{t=1}^{T}-\frac{1}{2\sigma_t^2}(\mathbf{v}_t - \mathbf{v}_q)^\top(\mathbf{v}_t - \mathbf{v}_q)\right)
\end{aligned}
\tag{1}
$$

where $\Sigma$ is a normalized term not related to $\mathbf{v}_q$. $F_{base}(\mathbf{v}_q)$ is maximized when $\frac{\partial F_{base}(\mathbf{v}_q)}{\partial \mathbf{v}_q} = 0$:

$$
-\frac{\hat{\mathbf{v}}_q - \boldsymbol{\mu}_q}{\sigma_q^2} + \sum_{t=1}^{T}\frac{\mathbf{v}_t - \hat{\mathbf{v}}_q}{\sigma_t^2} = 0
\tag{2}
$$

Hence

$$
\hat{\mathbf{v}}_q = \frac{\frac{1}{\sigma_q^2}\boldsymbol{\mu}_q + \sum_{t=1}^{T}\frac{1}{\sigma_t^2}\mathbf{v}_t}{\frac{1}{\sigma_q^2} + \sum_{t=1}^{T}\frac{1}{\sigma_t^2}}
\tag{3}
$$

## 2.2 KFAtt-freq

To estimate the hidden variable $\mathbf{v}_q$ with a frequency capping mechanism, the function to be maximized in KFAtt-freq is given by:

$$F_{freq}(\mathbf{v}_q, \mathbf{v}_{m=1:M}) = \varphi(\mathbf{v}_q | \boldsymbol{\mu}_q, \sigma_q^2 I) \prod_{m=1}^{M} \left[ \varphi(\mathbf{v}_m | \mathbf{v}_q, \sigma_m^2 I) \prod_{t=1}^{n_m} \varphi(\mathbf{v}_{m,t} | \mathbf{v}_m, \sigma_m'^2 I) \right]$$

$$= \frac{1}{\Sigma} \exp \left( -\frac{1}{2\sigma_q^2} (\mathbf{v}_q - \boldsymbol{\mu}_q)^\top (\mathbf{v}_q - \boldsymbol{\mu}_q) \right. \tag{4}$$

$$\left. + \sum_{m=1}^{M} \left[ -\frac{1}{2\sigma_m^2} (\mathbf{v}_m - \mathbf{v}_q)^\top (\mathbf{v}_m - \mathbf{v}_q) + \sum_{t=1}^{n_m} -\frac{1}{2\sigma_m'^2} (\mathbf{v}_{m,t} - \mathbf{v}_m)^\top (\mathbf{v}_{m,t} - \mathbf{v}_m) \right] \right)$$

where $\Sigma$ is a normalized term not related to $\mathbf{v}_q$ and $\mathbf{v}_m$. $F_{freq}(\mathbf{v}_q, \mathbf{v}_{m=1:M})$ is maximized when $\frac{\partial F_{freq}}{\partial \mathbf{v}_q} = 0$ and $\frac{\partial F_{freq}}{\partial \mathbf{v}_m} = 0$:

$$-\frac{\hat{\mathbf{v}}_q - \boldsymbol{\mu}_q}{\sigma_q^2} + \sum_{m=1}^{M} \frac{\hat{\mathbf{v}}_m - \hat{\mathbf{v}}_q}{\sigma_m^2} = 0 \tag{5}$$

$$-\frac{\hat{\mathbf{v}}_m - \hat{\mathbf{v}}_q}{\sigma_m^2} + \sum_{t=1}^{n_m} \frac{\mathbf{v}_{m,t} - \hat{\mathbf{v}}_m}{\sigma_m'^2} = 0, \forall m \in 1 \dots M \tag{6}$$

Hence

$$\hat{\mathbf{v}}_q = \frac{\frac{1}{\sigma_q^2} \boldsymbol{\mu}_q + \sum_{m=1}^{M} \frac{1}{\sigma_m^2} \hat{\mathbf{v}}_m}{\frac{1}{\sigma_q^2} + \sum_{m=1}^{M} \frac{1}{\sigma_m^2}} \tag{7}$$

$$\hat{\mathbf{v}}_m = \frac{\frac{1}{\sigma_m^2} \hat{\mathbf{v}}_q + \frac{n_m}{\sigma_m'^2} \overline{\mathbf{v}}_m}{\frac{1}{\sigma_m^2} + \frac{n_m}{\sigma_m'^2}} \tag{8}$$

where $\overline{\mathbf{v}}_m = \frac{1}{n_m} \sum_{t=1}^{n_m} \mathbf{v}_{m,t}$. Substituting $\hat{\mathbf{v}}_m$ into Eq (7) we obtain

$$\hat{\mathbf{v}}_q = \frac{\frac{1}{\sigma_q^2} \boldsymbol{\mu}_q + \sum_{m=1}^{M} \frac{1}{\sigma_m^2} \frac{\frac{1}{\sigma_m^2} \hat{\mathbf{v}}_q + \frac{n_m}{\sigma_m'^2} \overline{\mathbf{v}}_m}{\frac{1}{\sigma_m^2} + \frac{n_m}{\sigma_m'^2}}}{\frac{1}{\sigma_q^2} + \sum_{m=1}^{M} \frac{1}{\sigma_m^2}} \tag{9}$$

Thus

$$\hat{\mathbf{v}}_q = \frac{\frac{1}{\sigma_q^2} \boldsymbol{\mu}_q + \sum_{m=1}^{M} \frac{1}{\sigma_m^2 + \sigma_m'^2 / n_m} \overline{\mathbf{v}}_m}{\frac{1}{\sigma_q^2} + \sum_{m=1}^{M} \frac{1}{\sigma_m^2 + \sigma_m'^2 / n_m}} \tag{10}$$

# 3 Statistics of Industrial Dataset

Table 2: JD's Real Production Dataset Statistics. Besides the features listed, we also do manual feature interaction, making the total number of features= 96.

| Field | # Features | #Vocabulary | Feature Example |
|---|---|---|---|
| **User Behaviors** | 1 | 300 M | clicked item id |
| Query | 4 | 20 M | query, brands in query, query segmentation |
| User Profiles | 6 | 400 M | user pin, location, price sensitivity |
| Ad Profiles | 17 | 20 M | ad id, category, item price, brands, ad title |
| Contexts | 4 | 70 | time, ad slot |

# 4 Additional Experiments

Table 3: Ablations studies of KFAtt.

| Data | Trans | KFAtt-bs | KFAtt-b | KFAtt-fs | KFAtt-f | KFAtt-f-Cate2 | KFAtt-f-Cate1 |
|------|-------|----------|---------|----------|---------|---------------|---------------|
| All | 0.8720 | 0.8740 | 0.8766 | 0.8754 | **0.8789** | 0.8775 | 0.8766 |
| New | 0.8488 | 0.8515 | 0.8552 | 0.8532 | **0.8578** | 0.8559 | 0.8556 |
| Infrq | 0.8414 | 0.8454 | 0.8465 | 0.8471 | 0.8496 | 0.8504 | **0.8506** |

We add this group of experiments (Table 3) to address the concerns from reviewers.

- The performance comparison to some naive and straightforward solutions that also include query-specific prior and frequency capping.
- KFAtt-freq's sensitivity to different deduplication algorithms.

First, we compare KFAtt-b (proposed in Section 3.2) to a naive solution KFAtt-bs, which simply adds a query-specific prior (using $\sigma_q = 1$). And we also compare KFAtt-f (proposed in Section 3.3) to a naive solution KFAtt-fs, which do simple frequency capping by neglecting $\sigma'_m$ in Eq 10. We find clear superior of the proposed algorithms to their naive counterparts. This validates that KFAtt is far more than 2 simple modifications but based on clear theoretical design. With the additional $\sigma_q$ and $\sigma'_m$, it assigns stronger prior and capping to specific queries than to general ones.

The Amazon dataset contains 3 levels of categories. KFAtt-f uses 3-rd level category for de-duplication. In comparison, we now show results when using 2nd and 1st level category for de-duplication. When comparing these two with KFAtt-f, we find that coarser de-duplications benefit queries from infrequent cates but harm frequent ones, leading to lower performance on All. In addition, KFAtt-f with any level of de-duplications outperforms KFAtt-b and other STOAs, which indicates that KFAtt-f is insensitive to deduplication algorithms.