[Reviews · NeurIPS 2020]

Review 1

Summary and Contributions: This paper proposes a new attention mechanism called Kalman Filter Attention (KFAtt) to solve two problems in CTR. One is that traditional attention mechanisms only focus on single user’s behaviors and the other is that frequent behaviors are overweight. Specifically, the authors consider the weighted pooling in attention as a maximum a posteriori (MAP) estimation to solve the first limitation, and propose a frequency capping mechanism to solve the second limitation. Offline experiments on a benchmark and a 10 billion scale real production dataset show the effectiveness of the KFAtt model.

Strengths: 1. The two limitations in traditional attention mechanisms raised by this paper are really important and general. Moreover, the author also gives some statistical data analyses to support the existence of these two limitations in Figure 1. Actually, it is reasonable to me. 2. The model proposed by the author is innovative and effective. Kalman filtering is indeed suitable for solving these two limitations. The KFAtt model can also solve the problem of cold start. 3. This paper has a good theoretical basis since it has a clear derivation of the proposed KFAtt model. 4. The KFAtt model is deployed in a 10 billion scale real production dataset, which is practical. And the experimental results show the effectiveness of the proposed model.

Weaknesses: 1 The related work section of this paper should be better organized. It needs more detailed description of the application of the attention mechanism in CTR prediction. For example, how does DIEN [1] apply attention mechanism into RNN? The main goal of this paper is to improve the attention mechanism, so I think the author should talk about the attention mechanism in related work systematically. [1]Guorui Zhou, Na Mou, Ying Fan, Qi Pi, Weijie Bian, Chang Zhou, Xiaoqiang Zhu, and Kun Gai. Deep interest evolution network for click-through rate prediction. In Proceedings of the AAAI Conference on Artificial Intelligence, volume 33, pages 5941–5948, 2019. 2. The experiment of this paper is somewhat not complete. For example, I find there is the result of DIEN model on Amazon Dataset, and however, the authors do not show its performance on production dataset. Could you explain why? It is necessary to give reasons. 3. This paper has a few grammatical errors. For example, on line 51, ‘degree’ should be ‘degrees’.

Correctness: this paper has a clear derivation of the proposed KFAtt model. The claims and methods are correct, and the empirical methodology sounds reasonable.

Clarity: The structure and logic of this paper are clear. However, the related work part of this paper should be better organized and needs more detailed description of the application of the attention mechanism in CTR prediction. Details can be found above

Relation to Prior Work: The Author introduces Kalman filtering into the attention mechanism and solve two limitations in previous attention methods. Generally, the contributions are clearly discussed.

Reproducibility: Yes

Additional Feedback:


Review 2

Summary and Contributions: This paper proposes a nonlocal attention mechanism, namely Kalman Filtering Attention, that aims to address the two common problems of existing attention mechanisms for being localized and vulnerable to query duplications. Apart from the offline evaluations, the authors have been able to deploy the developed method on a large-scale e-commerce platform, which is beneficial for showcasing its real-life applicability.

Strengths: 1. The observations on existing attention mechanisms’ shortcomings are sensible. The examples in Figure 1 also depicts the necessity of capturing nonlocal user interest patterns and offsetting the impact from redundant interactions from a global view. 2. The proposed KFAtt seems extendable to different neural architectures. It employs other users’ interest estimated via MLPs, where the idea of modelling the deduplicated query sequence is simple yet effective for reducing the noise from popular queries. 3. The authors performed online A/B tests on top of offline experiments, which demonstrates KFAtts’ performance gain and scalability on industry-level datasets.

Weaknesses: 1. While there is a claim that KFAtt can predict users’ hidden interests with the global attribute even when few historical behaviors are relevant (page 2 line 53), I do not see sufficient experimental analysis that can support this. The authors may resolve this by introducing a hyperparameter study which adjusts the lengths of historical records used for prediction. 2. The implementation details indicate that the CTR prediction is actually conducted in a session-based manner. In session-based recommendation where user identity is untraceable, it might not be very reasonable to utilize the user’s long history records, e.g., previous 70 days’ click records (page 6 line 233) as input features. Furthermore, it defeats the purpose of the global attribute which is designed to provide nonlocal user interest information. 3. It should be clarified that how an accurate estimation of \mu_q, \sigma_q, and \sigma_m’ can be guaranteed from a 2-layer MLP with only p/k as the input. Are there corresponding ground truth labels available during training?

Correctness: The proposed method is correct.

Clarity: The writing quality is good. However this paper misses a few details that could have made it easier to follow, e.g., the semantics of \mu_q – is that an average pooling of all users’ embeddings? And what kind of distance measure is used to quantify the value of \theta_t using k_t and q?

Relation to Prior Work: The discussion on prior work mainly concerns different variants of attention nets. However, this paper should also discuss and compare with some mainstream CTR models (especially Factorization Machine-based models) in the literature, e.g., xDeepFM (KDD’18), AFM (IJCAI’17), etc.

Reproducibility: No

Additional Feedback: Some notes on reproducibility and experiments: The model itself is not hard to implement but I am expecting a comprehensive list of all hyperparameter settings (e.g., number of attention heads, the optimizer, any regularization strategies, etc.) to ensure the reproducibility regarding the reported results. An open-source link would help. Also, apart from AUC, I recommend reporting LogLoss as well because it is also an important metric for estimating business revenue in the industry. Some edits: Page 2 line 45, “close-related” --> closely related Page 2 line 72, “theoretical analyse” --> theoretical analysis Page 2 line 112, “can not” --> cannot


Review 3

Summary and Contributions: The paper proposes a modifying a standard attention mechanism to tackle two problems that arise in CTR: (a) for new items not seen in the context, a query derived attention weight and value (in the standard QKV attention parlance); (b) to reduce the double-counting of repeated keys, treating key-value pairs under the same key as a single key and use the averaged value. Experiment on Amazon dataset and a proprietary CTR system, and find statistically significant improvements.

Strengths: * Practical improvements to attention mechanisms to tackle real problems in CTR systems. * Empirical improvements on an academic dataset real world (proprietary) experimental setup.

Weaknesses: * It's not clear how the Kalman Filtering perspective provides any new insight. Both the global query-specific prior and frequency capping are trivial to specify in the standard attention framework. The Kalman Filtering perspective seems like an unnecessary distraction from what is in reality two simple modifications to a standard attention mechanism. It's especially confusing since a Kalman filter is traditionally specified as a sequential mechanism, more similar to an RNN than a Transformer. Section 3.5 doesn't sufficiently address these questions. For example, is the difference between expectation and estimation important in this setting? I currently view the KF perspective as a negative that distracts from the core modeling changes. * There are only two benchmark results. While the improvements are statistically significant, it's unclear whether they are nontrivial improvements. The authors need to provide more context here, especially for the real-world system. For example, is a +4.4% CTRgain big or small for this system?

Correctness: Yes.

Clarity: Overall, it's well written. Context is made clear even for readers without a background in CTR modeling. However, the final *implemented* mechanism is hard to immediately tell from reading the manuscript, since different implementation choices for each component (e.g., \mu_q) are spread across the paper. Could this be clarified in the rebuttal? Furthermore, if exp(q^T W_Q W_K k_m) is used as 1/sigma_m^2, how is the KFAtt-f calculated? In particular, if \prime{sigma_m^2} is calculated with an MLP, how are these two quantities combined? With all the reciprocals, it seems that a naive implementation would be numerically unstable. How does it work?

Relation to Prior Work: Yes.

Reproducibility: No

Additional Feedback: Overall, I think the paper holds promise. The problems outlined in the work seem general across user modeling systems. The results, if the authors can give more context, may be significant, which is important given the predominance of CTR systems in e-commerce and advertising. I'm willing to reconsider my score if in the rebuttal, the authors (a) explain why the KF perspective is useful; (b) provide more context about the significance of the real world system results.


Review 4

Summary and Contributions: This paper introduces a modified attention mechanism that allows 1) attention on a global prior for a particular query, and 2) attention to popular classes of queries to be capped in a principled manner. The mechanism is derived by analogy to kalman filters, with attention weights expressed as the MAP estimate given "measurements" corresponding to prior queries (plus the global prior). The empirical analysis includes both public data and large-scale e-commerce logs, as well as live experiments on a high-traffic system.

Strengths: The paper is well presented, and the problem it addresses is well motivated. The empirical study has good scale and covers open reproducible data as well as an interesting analysis on proprietary data that includes live-traffic A/B experiments. I find the results in Table 3 compelling, as this seems to be good evidence of lift from all components in a mature system. However, it would be nice to see some representation of error bars / variation in these results.

Weaknesses: The incorporation of the global prior as one element of attention seems equivalent to including this as a "special" entry in the history -- as far as I can tell it is treated identically in Eq 6 and 10. Hence, the performance should be identical to a residual learned against the global prediction, as the relative contribution of the remaining portions of attention can be learned with respect to the implicit weight of 1 given to the prior. I'm therefore not sure that the global information is a novel contribution of the work. The derivation in Eq 10 of an appropriate weighting for the attention to a cluster of previous queries provides some principle for the form, which is nice. However, there could be a lot of sensitivity to the specifics of the de-duplication algorithm. I did not see any analysis of this. Regarding the analysis in Table 2, and the corresponding discussion in Section 4.5 (line 270): it seemed that the improvements here when swapping in the values learned by KFAtt into the Vanilla and other models could be viewed as an error in the hyperparameter search of those systems, or a shortcoming in the tuning process. I enjoyed the paper -- the primary weakness I see is that the level of novelty in the new mechanisms is a little low. I appreciate the authors' response, and I remain supportive of this paper based on it.

Correctness: Yes, to my knowledge.

Clarity: Yes, the best-written one in my batch.

Relation to Prior Work: Yes, it seemed adequate to me.

Reproducibility: Yes

Additional Feedback:

[Author Response · NeurIPS 2020]

We thank all the reviewers for their valueable suggestions. Typos will be fixed and related works will be revised.

**R1Q1**: No DIEN results on product dataset. Necessary to give reasons.

**R1A1**: In preliminary studies, DIEN performs worse than Transformer on product dataset (consists with Tab1). We did
not report the exact number because DIEN is not feasible given the high latency in our CPU-based online system (Fig2).

**R2Q1**: Claim "KFAtt predict interests when few historical behaviors are relevant, Line53" lacks experimental support.

**R2A1**: Line256-258&264 and Tab1&2. We construct a more challenging test subset "*New*", where *NO* historical
behaviors are relevant to the current query. On "New", KFAtt achieves *larger* performance gain compared to that on
"All", directly supporting the claim. A study of history length is helpful, but not essential given results on "New".

**R2Q2**: The implementation details are in a session-based CTR prediction, not reasonable to use long history.

**R2A2**: Our system is NOT in *session-based* CTR prediction. Long behaviors of identities are totally traced. Our
problem setting is *user behavior modeling* [23,24,25] (*Our Title*), i.e., to predict user's current interest from rich
historical behaviors, Line26. We apologize for that this misunderstanding might come from a reference compiling error.
Line189 [11] should be *Feng Yufei. Deep session interest network for click-through rate prediction.* "Session" here
means a partition method in processing very long behavior sequences, irrelevant to "Session"-based CTR prediction.

**R2Q3**: How an accurate estimation of $\mu_q$, $\sigma_q$ $\sigma'_m$ from a 2-layer MLP with only p/k as input? Ground truth available?

**R2A3**: Only embeddings $\mathbf{q}$, $\mathbf{k}$ are needed for $\mu_q, \sigma_q, \sigma'_m$ (Line127&159). No ground truth. The intuitive reason for this
simple but accurate estimation is that they are trained and shared across a great many users with the same query.

**R2Q4**: The semantics of $\mu_q$? And what kind of distance measure is used to quantify the value of $\theta_t$ using $k_t$ and $q$?

**R2A4**: $\mu_q$ is Gaussian distribution mean, namely the mean interest under same $\mathbf{q}$ across all users (Line122-124), NOT
avg-pool of user embedding. Calculated by $\mu_q = MLP(\mathbf{q})$ (Line127). Distance measured as Line200.

**R2Q5**: Reproducibility. **R2A5**: #heads=4. Others follow codes of DIEN. Will consider open source upon acceptance.

**R3Q1**: How Kalman Filtering (KF) provides new insights, given the simple solutions to the two problems?

**R3A1**: Although useful in sequential scenarios, KF is essentially a *sensor-fusion* method. The fusion is estimated by
MAP, whose solution is a weighted-sum of *prior* and sensor measurements. Similarly in behavior modeling, each
historical behavior can be considered as a measurement of the current interest. So the current interest is also a fusion,
which is naturally under MAP framework and thus fits KF. While conventional attentions (*expectation*) neglect query
priors and thus suffer from cold start. Moreover, KFAtt is far more than "2 simple modifications". With the additional
$\sigma_q$ and $\sigma'_m$, it assigns stronger prior and capping to specific queries than to general ones, Line125. To see this superior,
we now show AUC gain from "simply including query-specific prior, $\sigma_q = 1$" to KFAtt-b, and gain from "simply
frequency capping" to KFAtt-f: *All*: +0.0026 | +0.0035  ⫼  *New*: +0.0037 | +0.0046  ⫼  *Infreq*: +0.0011 | +0.0025.

**R3Q2**: Whether it is nontrivial improvements. Is a +4.4% CTR gain big or small?

**R3A2**: Our base is highly optimized (400M users Tab2 Supp), CTR+4.4% => Income+\$0.1Billion/year, Big Gain.

**R3Q3**: Different implementation choices (e.g. $\mu_q$) spread across the paper.

**R3A3**: Query mean and std ($\mu_q, \sigma_q, \sigma'_m$) are from MLP (Line127&159). Distance $\sigma_t, \sigma_m$ are as Line200. This is the
only final implementation. Other variants are only for ablation studies of adaption to other attentions (Tab2).

**R3Q4**: If $\exp(\mathbf{q}^T W_Q W_K \mathbf{k}_m)$ is used as $1/\sigma_m^2$, how is KFAtt-f calculated? Numerically stable?
**R3A4**: In Eq(10), $\frac{1}{\sigma_m^2 + \sigma_m'^2/n_m} = \frac{1/\sigma_m^2}{1 + 1/\sigma_m^2 \cdot \sigma_m'^2/n_m} = \frac{\exp(\mathbf{q}^\top W_Q W_K \mathbf{k}_m)}{1 + \exp(\mathbf{q}^\top W_Q W_K \mathbf{k}_m) MLP(\mathbf{k}_m)/n_m}$. Numerically stable.

**R4Q1**: Is global prior equivalent to including an "entry" weighed by 1?    |    **R4A1**: No. Pls see R3A1 for details.

**R4Q2**: Sensitivity analysis to de-duplication algorithm.

**R4A2**: Amazon dataset contains 3 levels of categories. We now show AUC gain from using 3rd level de-duplication (as
in paper) to 2nd, and gain from 3rd to 1st. *All*: -0.0014 | -0.0023  ⫼  *New*: -0.0019 | -0.0022  ⫼  *Infreq*: +0.0008
| +0.0010. Coarser de-duplications benefit queries from infrequent cates but harm frequent ones, leading to lower
performance on *All*. KFAtt-f with any level of de-duplications outperforms KFAtt-b and other STOAs, not that sensitive.

**R4Q3**: Swapping the values learned by KFAtt to Vanilla and other models is wrong.

**R4A3**: Thank reviewer's help in finding an ambiguous description. Line272 should be "we assign attention weights
calculated by Vanilla, DIN and Transformer to $1/\sigma_t^2$ and $1/\sigma_m^2$ in Eq (6,10)". Namely, we plug KFAtt to these attentions
and show consistent improvements brought to them.

[Meta-Review · NeurIPS 2020]

All reviewers are positive of this papers contributions in both methodology and experiments. Hence I am happy to suggest acceptance. Couple of comments about the paper that I hope authors will address in the final version. 1) Some of the drawbacks/ideas about user modeling have been explored thoroughly in the collaborative filtering/matrix completion setup. However this paper currently lacks such references. Drawing connections to this field of related work is important. 2) The paper can benefit from including more experiments on publicly available datasets.